# Comparative analysis of WC1.1[+] and WC1.2[+] γδ T cell subset responses from cattle naturally infected with *Mycobacterium bovis* to repeat stimulation with mycobacterial antigens

**Alia Parveen**[1☉], **Sajad A. Bhat**[1,2,3☉], **Mahmoud Elnaggar**[4,5☉], **Kieran G. Meade**[1,2,6]*

**1** UCD School of Agriculture and Food Science, University College Dublin, Dublin, Ireland, **2** UCD Conway Institute of Biomolecular and Biomedical Research, University College Dublin, Dublin, Ireland, **3** Animal and Bioscience Research Department, Animal and Grassland Research and Innovation Centre, Teagasc, Grange, Meath, Ireland, **4** Department of Veterinary Medicine, College of Applied and Health Sciences, A'Sharqiyah University, Ibra, Oman, **5** Department of Microbiology, Faculty of Veterinary Medicine, Alexandria University, Egypt, **6** UCD Institute of Food and Health, University College Dublin, Dublin, Ireland

☉ These authors contributed equally to this work.
* kieran.meade@ucd.ie

**Data Availability Statement:** All RNA-seq data generated for this study have been deposited in the

## Abstract

*Mycobacterium bovis* (*M. bovis*) causes bovine tuberculosis (bTB). The challenges in controlling and eradicating this zoonotic disease are compounded by our incomplete understanding of the host immune response. In this study, we used high-throughput bulk RNA sequencing (RNA-seq) to characterise the response profiles of γδ T cells to antigenic stimulation using purified protein derivate from *M. bovis* (PPDb). γδ T cells are a subgroup of T cells that bridge innate and adaptive immunity and have known anti-mycobacterial response mechanisms. These cells are usually classified based on the expression of a pathogen-recognition receptor, Workshop Cluster 1 (WC1), into two main subsets: WC1.1[+] and WC1.2[+]. Previous studies have identified a preferential transcriptomic response in WC1.1[+] cells during natural bTB infection, suggesting a subset-specific response to mycobacterial antigens. This follow on study tested the hypothesis that a subset specific response would also be apparent from γδ T cells from infected cattle after repeat stimulation. Peripheral blood was collected from Holstein-Friesian cattle naturally infected with *M. bovis*, confirmed by a single intradermal comparative tuberculin test (SICTT) and IFN-γ ELISA and stimulated with 10 μg/ml PPDb for 6 hours. After whole blood stimulation, WC1.1[+] and WC1.2[+] γδ T cell subsets were isolated using magnetic cell sorting (n = 5 per group). High-quality RNA was extracted from each purified lymphocyte subset (WC1.1[+] and WC1.2[+]) to generate transcriptomes using bulk RNA sequencing, resulting in 20 RNA-seq libraries. Transcriptomic analysis revealed 111 differentially expressed genes (DEGs) common to both WC1.1[+] and WC1.2[+] γδ T cell compartments, including upregulation of *IL1A*, *IL1B*, *IL6*, *IL17A*, *IL17F*, and *IFNG* genes (FDR-$P_{adj.}$ < 0.1). Interestingly, the WC1.2[+] cells showed upregulation of *IL10*, *CCL22*, and *GZMA* (log$_2$FC ≥ 1.5, and FDR-$P_{adj.}$ < 0.1). In conclusion, while WC1.1[+] and WC1.2[+] γδ T cells exhibit a conserved inflammatory response to PPDb, differences in

ArrayExpress database under the project accession number E-MTAB-14366.

**Funding:** This project was funded by Science Foundation Ireland (SFI) Grant 17/CDA/4717 and AP was funded by Higher Education Authority (HEA) to Vit-TB. The funders had no role in study design, data collection and analysis, decision to publish, or preparation of the manuscript.

**Competing interests:** The authors have declared that no competing interests exist.

anti-inflammatory and antimicrobial gene expression between these cell subsets provide new insights into their effector functions in response to mycobacterial antigens.

## Introduction

Many mycobacterial species, including *Mycobacterium avium* subsp. *paratuberculosis* (MAP) and *Mycobacterium bovis* (*M. bovis*) have significant economic and animal welfare impacts on the global livestock industry. MAP is responsible for causing a chronic wasting gastroenteritis called Johne's disease [1], and *M. bovis* is the cause of bovine tuberculosis (bTB), which also poses a significant threat to public health as a zoonotic pathogen [2, 3]. Despite extensive efforts for eradication, bTB remains a challenge in many countries, including Ireland, due to various factors, including an incomplete understanding of *M. bovis* persistence in natural infection settings [4]. Current disease control programmes rely on a single or combination detection of cell-mediated immune responses known as the *Single Intradermal Comparative Tuberculin Test (SICTT)*, an *in vivo* skin test that measures the activation of a Delayed-Type Hypersensitivity (DTH) reaction [5] within 72 hours after the administration of both bovine and avian mycobacterial tuberculin antigens (*M. bovis/M. avium* purified protein derivative–PPDb/PPDa) and the *interferon-gamma assay* (IFN-γ) is often used as an ancillary test, interpreted in parallel with the SICTT to improve the sensitivity of diagnostic testing. The principle of the IFN-γ assay is to detect and quantify the release of the IFN-γ cytokine when heparinised whole blood is incubated with PPDb and PPDa [6]. However, despite annual testing, a reservoir of infection exists [7], which frustrates successful eradication efforts. In that context, an improved understanding of the immune response in bTB infected cattle, as well as the immune capacity of effector cells in such cattle, could identify how *M. bovis* persistence is influenced by the intricate host-pathogen interactions occurring during natural infection.

γδ T cells, as their name suggests, are a subset of T lymphocytes which differ from their conventional and usually more numerous αβ T cell counterparts using a different TCR configuration as well as pathogen recognition receptors (PRRs) capable of direct activation [8, 9]. Coupled with the route of activation via antigen presentation to the TCR, γδ T cells, therefore, occupy a niche which spans both innate and adaptive immunity. These cells have attracted significant interest in both human and murine studies due to their lineage-specific development of cell subsets which confers an ability to secrete specific effector cytokines that shape the consequential immune response [10]. Two divergent models have been proposed to explain the origin and functional specialisation of these cells: the predetermination model, which suggests that effector fate is determined prior to TCR expression, and the TCR-dependence model, which proposes that the nature of the TCR signal, particularly its intensity or duration, plays an important role in influencing effector fate. This thesis remains unresolved with one author describing these cells as "a riddle wrapped in an enigma" [9], which is further complicated by species-specific differences in γδ T cells development and function. A particularly relevant evolutionary facet of γδ T cells is their numeric expansion in the cattle lineage where they can account for up to 60% of circulating lymphocytes in contrast to approximately 5% in humans and mice [11, 12]. The reason for this species (or clade)-specific expansion remains unknown but may be related to the regulation of the complex microbial niches in livestock. What has been shown is that γδ T cells are the first to develop post-natally and they then subsequently migrate to the periphery, where they play critical roles across mucosal epithelial surfaces [13]. However, much remains to be discovered about the functions of these mysterious cells, particularly in the context of disease.

Bovine γδ T cells express a pathogen recognition receptor known as workshop cluster 1 (WC1) receptor, dividing them into WC1[+] and WC1[-] subsets, further classified into WC1.1[+] and WC1.2[+] populations [14, 15]. These subsets exhibit diverse functionalities, including proinflammatory and anti-inflammatory functions, thereby influencing cattle's ability to mount effective anti-mycobacterial immune responses against *M. bovis* [16, 17], including the secretion of the bTB test-relevant cytokine, IFN-γ [18, 19], and these cells can thereby also influence the activity of other key innate cells including dendritic cells and macrophages [20–22]. One study demonstrated that circulating bovine γδ T cells secrete the anti-inflammatory cytokine IL-10 which inhibits the proliferation of both CD4[+] and CD8[+] T cells, thereby documenting a major regulatory role for these cells [23]. Despite their critical role, studies comprehensively assessing bovine γδ T cells under natural bTB infection conditions are scarce. Our previous analysis identified significant differential gene expression in the WC1.1[+] γδ T cells from cattle naturally infected with *M. bovis*-infected compared to control groups [24], and these included genes encoding important functional proteins such as granzymes with cytotoxic potential. Both WC1.1[+] and WC1.2[+] cell subsets are known to respond to mycobacterial antigens [16–19], but the commonalities and subset-specific modalities of the cell subsets have not been comprehensively defined. In this study, using RNA-seq, we delve deeper into the response of WC1.1[+] and WC1.2[+] subpopulations from bTB-positive (bTB+) cattle to PPDb stimulation, aiming to unravel the specific roles of these subsets in the primary and secondary immune response to *M. bovis* antigen challenge.

## Materials and methods

### Experimental animals

Male Holstein-Friesian (*B. taurus*) cattle were used for this study. Animals (n = 5) were selected from a herd of animals naturally infected with *M. bovis* maintained at the Department of Agriculture, Food and Marine research farm in Longtown, Co. Kildare, Ireland. The infected animals had tested positive for bovine tuberculosis by single intradermal comparative tuberculin test (SICTT) and also by the whole blood interferon (IFN)-γ release assay (IGRA, University College Dublin, Ireland). All animal procedures and experimental protocols in this study were approved by the Teagasc Animal Ethics Committee (TAEC no. TAEC217-2019) and carried out in strict accordance with the relevant institutional guidelines and under license from the Irish Health Products Regulatory Authority (HPRA no. AE19132/I019).

### Blood sampling, stimulation, γδ T cell labelling and sorting

Peripheral blood from the jugular vein of bTB infected animals was collected in 9 ml vacutainer tubes containing sodium heparin anticoagulant. The whole blood was stimulated with bovine purified protein derivative (PPDb) (10 μg/ml) for 6 hours in a humidified atmosphere with 5% $CO_2$ at 37°C. Non-stimulated whole blood cultures were also included as controls. At the end of incubation, PBMCs from each culture were harvested by density gradient centrifugation using Histopaque (Density 1.077 g/mL; Sigma-Aldrich). PBMCs were washed twice in PBS and then subjected to magnetic cell separation of γδ T cells, WC1.1[+] and WC1.2[+] subsets using primary monoclonal antibodies (mAbs) from Washington State University Monoclonal Antibody Center (WSUMAC), Pullman, WA, USA and anti-mouse IgG magnetic microbeads (Miltenyi Biotec, UK) according to manufacturer protocols. The primary mAbs used in this study were GB21A, an IgG2b isotype with catalogue number BOV2058, which is specific to the γδ TCR. For the WC1.1[+] marker, the mAb BAQ89A, an IgG1 isotype with catalogue number BOV2055, was utilised. The mAb CACTB32A, an IgG1 isotype with catalogue number

BOV2054, was also used to target the WC1.2$^+$ marker. Isolated cell samples were washed in PBS, pelleted in microfuge tubes, and stored at -80˚C.

## Total RNA extraction and library preparation

Total RNA was extracted from the cell pellets using AllPrep DNA/RNA Mini Kit (Qiagen) per the manufacturer's instructions. RNA quantity, integrity, and purity were assessed using a NanoDrop™ 1000 spectrophotometer (Thermo Fisher Scientific) and an Agilent 2100 Bioanalyzer using an RNA 6000 Nano LabChip kit (Agilent Technologies Ltd., Cork, Ireland), according to the manufacturer's instructions. All RNA samples used for transcriptomic analysis ($n$ = 5) had RNA integrity number (RIN) values >7.5. RNA libraries were prepared from a starting quantity of 150 ng high-quality RNA sequenced at Eurofins Genomics (Constance, Germany) using an INVIEW Transcriptome Discover product. This included purification of mRNA, fragmentation, strand-specific cDNA synthesis, end-repair, ligation of sequencing adapters, amplification and purification. The prepared libraries were then quality-checked, pooled and sequenced on an Illumina platform to generate 2 × 150 paired-end reads (Illumina NovaSeq6000, PE150 mode). All RNA-seq data generated for this study have been deposited in the ArrayExpress database under the project accession number E-MTAB-14366.

## RNA-seq data processing and analysis

Read quality was assessed using FastQC (version 0.11.9). Following quality analysis, reads were trimmed to remove low-quality and adapter sequences using Trimmomatic (version 0.39). FastQC analysis of trimmed-read quality analysis indicated that further read filtering and trimming was not required. Paired-end reads were aligned to the most recently annotated version of the *B. taurus* reference genome (ARS-UCD1.2) obtained from Ensembl [25] using the STAR RNA-seq aligner package (version 2.7.10). On average, 95.5% of paired-end reads were uniquely aligned, and 2.5% of read pairs were mapped to multiple loci. The number of aligned reads mapping to each gene annotated in the *B. taurus* genome was determined for each library using featureCounts from the Subread package (version 3.22.0). On average, 82% of aligned read pairs could be assigned to an annotation feature. *P* values were adjusted for multiple testing using the Benjamini-Hochberg (B-H) false discovery rate (FDR) method. The criteria for detection of significantly differentially expressed genes (DEGs) were an FDR-adjusted *P*-value less than 0.1 (FDR-$P_{adj.}$ < 0.10) and a $|\log_2\text{FC}| \geq 1.5$, which was incorporated into the statistical model in DESeq2.

Comparisons were made between WC1.1$^+$ stimulated versus unstimulated, WC1.2$^+$ stimulated versus unstimulated, and WC1.1$^+$ stimulated versus WC1.2$^+$ stimulated. Genes with an FDR value ($P_{adj.}$) of less than 0.1 were considered differentially expressed.

## GO and KEGG enrichment analyses

Lists of genes with increased or decreased expression $|\log_2\text{FC}| \geq 1.5$ and FDR-$P_{adj.}$ < 0.10) obtained through differential gene expression analysis were analysed. Enriched GO biological processes within the *B. taurus* gene database by performing GO enrichment analysis using the clusterProfiler package (version 4.8.3) in R. The differentially expressed gene list was analysed, and significantly enriched GO terms were identified with a *p* and *q value* cutoff <0.1. Similarly, KEGG pathway enrichment analysis was conducted using the clusterProfiler package (version 4.8.3) in R. The differentially expressed gene list was analysed, and significantly enriched KEGG pathways were identified with a *q value* (FDR-$P_{adj.}$ < 0.10). The *q value* is a common metric in a KEGG enrichment analysis and is a false discovery rate (FDR) adjusted

*P*-value accounting for multiple testing. Results were interpreted in the context of bovine biology, and visualisations were generated using ggplot2 (version 3.4.4) and other relevant tools.

## Results and discussion

### Clear activation of WC1.1$^+$ and WC1.2$^+$ γδ T cell subsets from bTB+ cattle in response to PPDb stimulation

Principal component analysis (PCA) was performed on the transcriptomic results from WC1.1$^+$ γδ T cells and WC1.2$^+$ γδ T cells isolated from whole blood of bTB+ cattle (n = 5) stimulated with PPDb and compared to non-stimulated cells (5 animals × γδ T cell subset × stimulation status = 20 RNA-seq libraries). The correlation coefficients of 20 sequenced samples show a clear divergence between PPDb stimulated and unstimulated WC1.1$^+$ and WC1.2$^+$ γδ T cell samples and PC1 and PC2 together explain 62% of the variance in transcriptomic responses (Fig 1).

A total of 912 genes were significantly differentially expressed in the WC1.1$^+$ γδ T cell subset in response to 6-hour PPDb stimulation FDR-$P_{adj.}$ < 0.10, reduced to 125 when a |log$_2$FC| ≥ 1.5 was applied (Fig 2A and 2B). Similarly, a total of 1214 genes were significantly differentially expressed in the WC1.2$^+$ γδ T cell subset in response to 6-hour PPDb stimulation FDR-$P_{adj.}$ < 0.10, reduced to 154 when a |log$_2$FC| ≥ 1.5 was applied (Fig 2A and 2B). The complete list of differentially expressed genes for each cell subset using the different stringency thresholds are provided in the S1 and S2 Tables, respectively.

The range in log$_2$FC exhibited was similar between subsets, varying between -1.73 and a maximum of 6.94 in WC1.1$^+$ and -1.6 to 7.56 in WC1.2$^+$. The vast majority of the significantly differentially expressed genes were increased in expression in both subsets (124 in WC1.1$^+$ and 152 in WC1.2$^+$), indicating a clear activation of a response to PPDb.

### Evidence for a conserved transcriptomic response in both WC1.1$^+$ and WC1.2$^+$ subsets to PPDb stimulation

Using a statistical cutoff of FDR-$P_{adj.}$ < 0.10, both WC1.1$^+$ and WC1.2$^+$ subsets showed an overlap of 800 of these genes, which were differentially expressed in common (Fig 3A). Enhancing the stringency of the comparison by imposing an additional |log$_2$FC| ≥ 1.5 reduces this number of genes to 111 (Fig 3B). All the differentially expressed genes in common between subsets were significantly increased in expression provided in the S3 Table.

The volcano plots in Fig 2 show multiple genes with well-established roles in the immune response were high fold change, significantly differentially expressed in response to PPDb and are represented in both WC1.1$^+$ and WC1.2$^+$ gene lists including *TLR4*, multiple cytokines including both isoforms of Interleukin 1 (*IL1A* and *IL1B*), Interleukin 2, Interleukin 6 and Interleukin 17 (*IL17A* and *IL17F*); chemokines including *CXCL8* and *CXCL10*; and other immune effector genes including *IRF1* and *ISG20*. Selected differentially expressed immune genes common to both WC1.1$^+$ and WC1.2$^+$ subsets are shown in Table 1, and the full list is shown in S3 Table.

Like other immune cells, the response of WC1$^+$ γδ T cell subsets to mycobacterial antigen involves recognition by pattern recognition receptors (PPRs), signal transduction, transcription factor (TFs) activation, and cytokine and chemokine production, leading ultimately to the initiation of adaptive immune responses. γδ T cells are optimally activated when both PPR co-receptors and TCRs are synergistically engaged despite being able to induce some level of activation with either one alone [26]. In this study, *TLR4* (Toll-like receptor 4) was upregulated by a log$_2$FC of 1.7 with FDR-$P_{adj.}$ = $1.08 \times 10^{-12}$ in WC1.1$^+$ and a log$_2$FC of 1.87 with FDR-$P_{adj}$ =

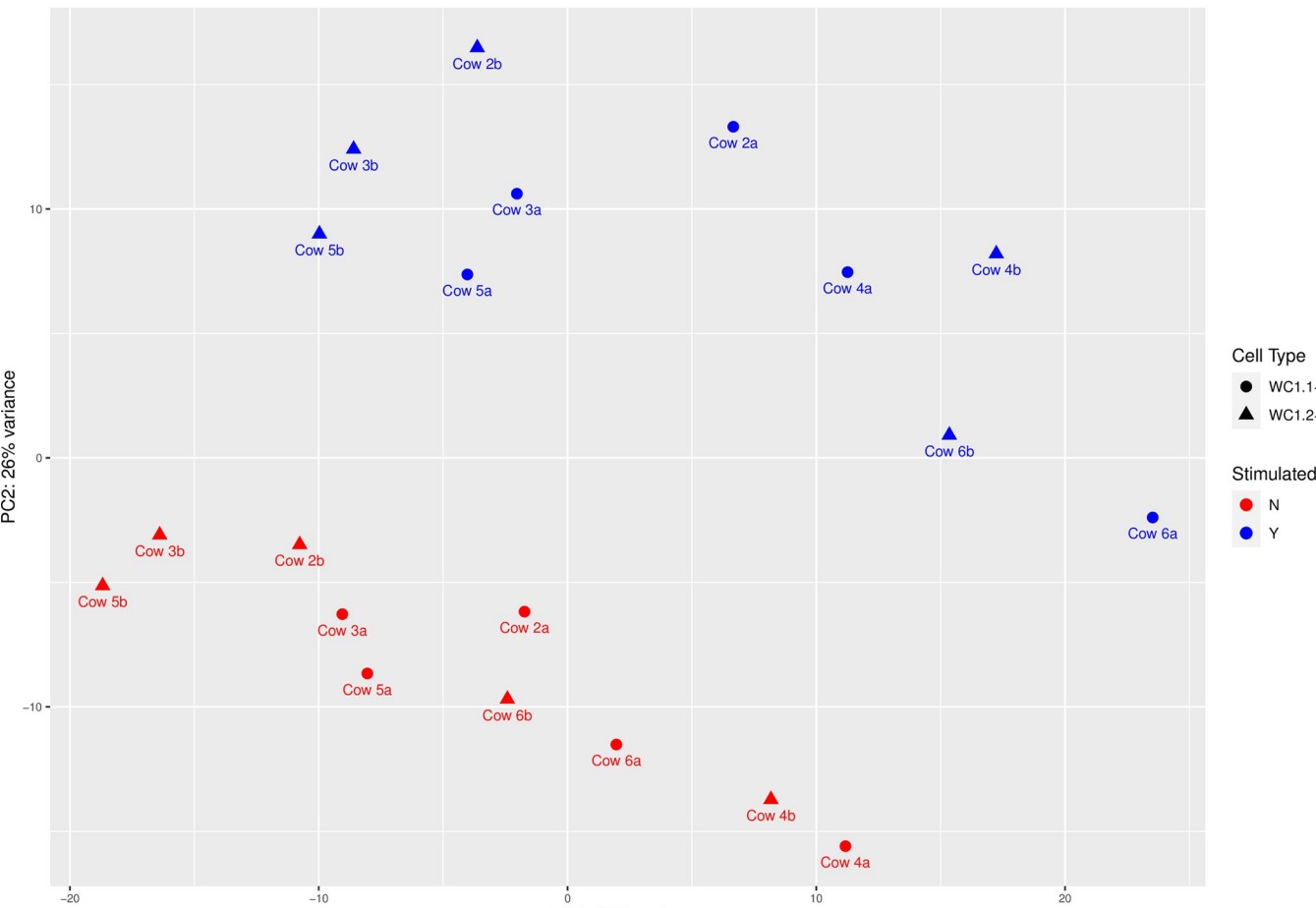

**Fig 1. Principal component analysis (PCA) of the transcriptomic expression from WC1.1$^+$ and WC1.2$^+$ γδ T cells from bTB+ cattle ($n$ = 5) in response to a 6-hour stimulation with 10 μg/ml PPDb compared to the non-stimulated.** Shape corresponds to γδ T cell type; the dot represents the WC1.1$^+$ subset, and the triangle represents the WC1.2$^+$ subset. Colour corresponds to 6-hour stimulation with 10 μg/ml PPD (blue) and non-stimulated (red).

$3.49 \times 10^{-15}$ in WC1.2$^+$ γδ T cell subsets. With PPDb representing a complex mixture of mycobacterial antigens and bTB+ cattle already being sensitised to mycobacteria, TLR4 is likely the pathogen recognition receptor protein involved in immune activation in both cell subtypes [27]. The gene encoding a soluble pattern recognition molecule, Pentraxin 3 (*PTX3*) also exhibits increased expression in both stimulated WC1.1$^+$ (log$_2$FC = 3.74, FDR-$P_{adj.}$ = $2.81 \times 10^{-8}$) and WC1.2$^+$ (log$_2$FC = 4.6, FDR-$P_{adj.}$ = $6.03 \times 10^{-12}$) cells, and is known to regulate the influx of leukocytes and possibly limiting inflammation-induced tissue damage [28].

Several downstream TFs, including *IRF1*, *IRF4*, and two members of the NF-κB family (*NFKB2* and *RELB*), showed increased expression in both stimulated γδ T cell subsets. In mammals, the interferon regulatory family of transcription factors (IRF1-9) have a role in T helper cell differentiation [29], and a previous study demonstrated extensive transcriptional programming in γδ T cells sorted from calves' peripheral blood mononuclear cells (PBMCs) following treatment with concanavalin A (ConA) [30]. *IRF1* (log$_2$FC in WC1.1$^+$ = 3.4 with FDR-$P_{adj.}$ = $3.91 \times 10^{-47}$; log$_2$FC in WC1.2$^+$ = 3.35 with FDR-$P_{adj.}$ = $1.18 \times 10^{-45}$) is critical for the IFN-γ signalling pathway, which enhances macrophage activation and Th1 responses which are essential for controlling *M. bovis* infection [31]. *IRF4* (log$_2$FC in WC1.1$^+$ = 2.14,

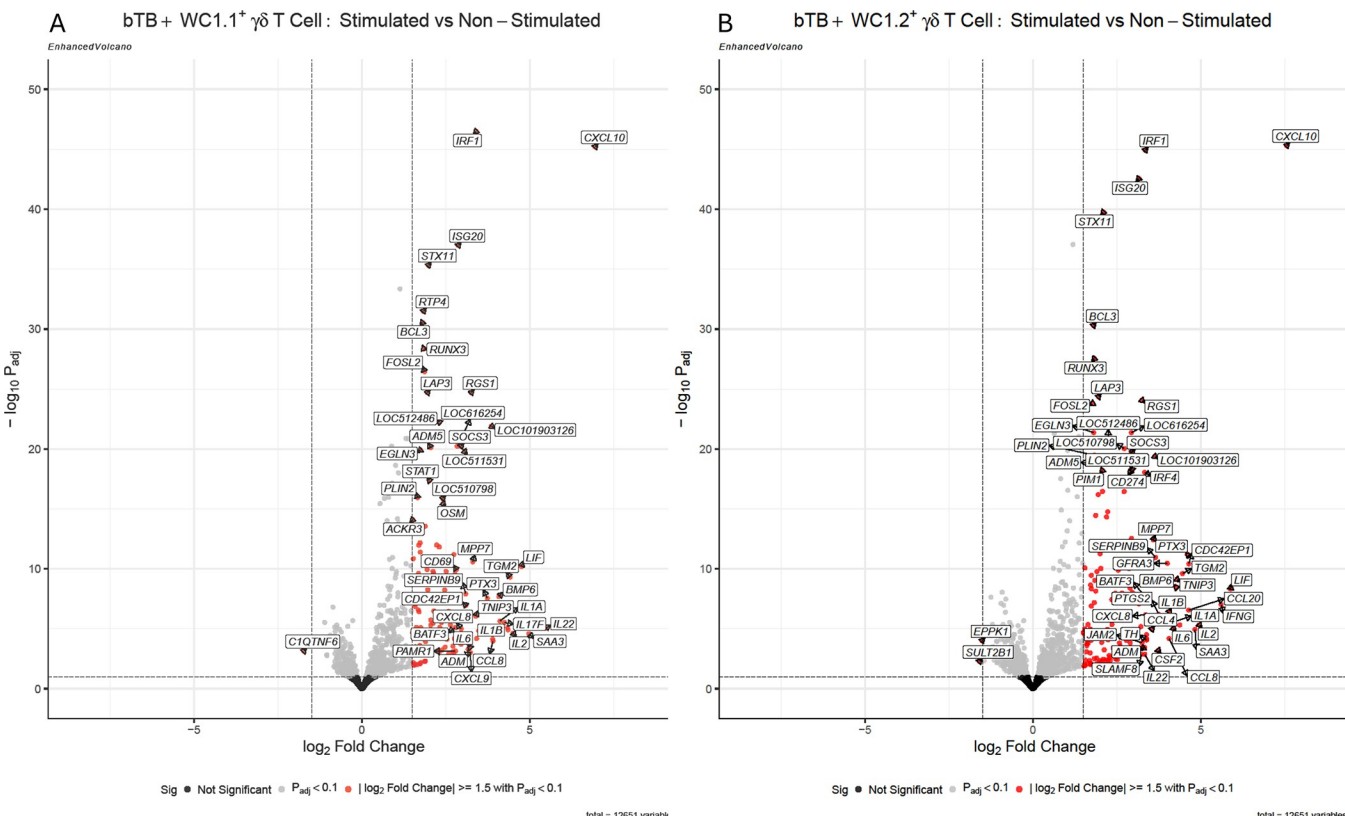

**Fig 2. Volcano plot for both upregulated and downregulated differentially expressed genes (DEGs) in response to a 6-hour stimulation with 10 μg/ml PPDb compared to the non-stimulated in (A) WC1.1⁺ γδ T cell subset and (B) WC1.2⁺ γδ T cell subset from bTB+ cattle ($n = 5$).** The red dots denote both up and down-regulated genes above the $|\log_2FC| \geq 1.5$ and FDR-$P_{adj.} < 0.10$ threshold (black dashed lines), grey dots denote DEGs with FDR-$P_{adj.} < 0.10$ only, and the black dots denote non-significant genes below the threshold ($\log_2FC| \geq 1.5$ and FDR-$P_{adj.} < 0.10$).

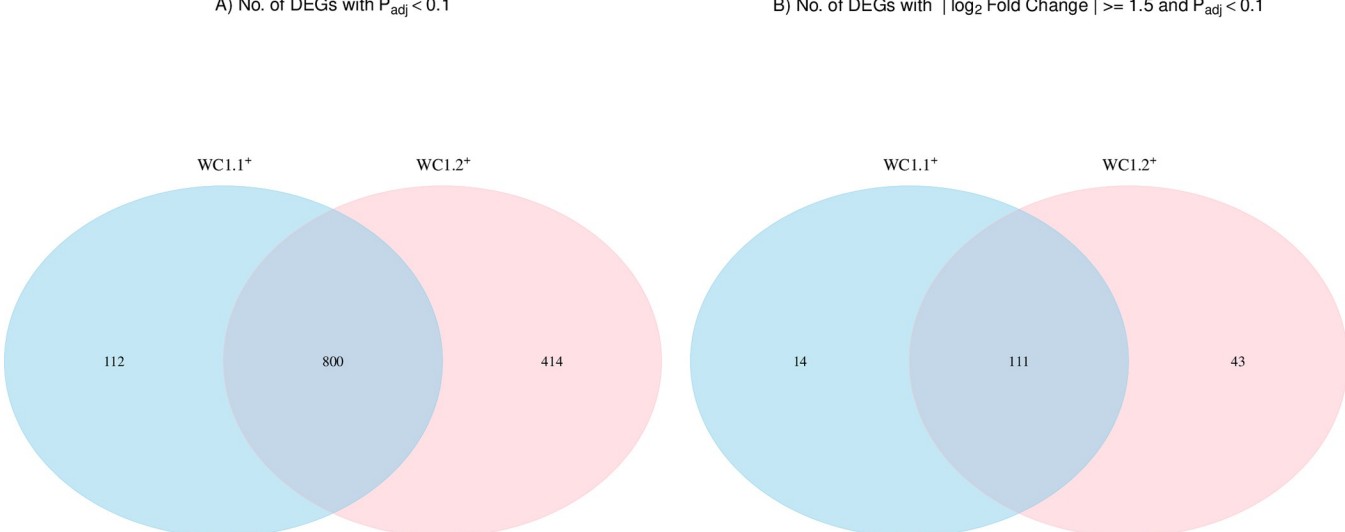

**Fig 3. Venn diagram showing the overlap of DEGs between WC1.1⁺ and WC1.2⁺ γδ T cell subsets from bTB+ cattle ($n = 5$) in response to a 6-hour stimulation with 10 μg/ml PPDb compared to the non-stimulated.** With A) FDR-$P_{adj.} < 0.10$ only and B) $|\log_2FC| \geq 1.5$ and FDR-$P_{adj.} < 0.10$.

**Table 1. Inflammatory genes exhibiting statistically significantly increased expression in WC1.1$^+$ and WC1.2$^+$ γδ T cell subsets (|log$_2$FC| ≥ 1.5, FDR-$P_{adj.}$ < 0.10) from bTB+ cattle ($n$ = 5) in response to a 6-hour stimulation with 10 μg/ml PPDb compared to the non-stimulated control.**

| Gene details | | WC1.1$^+$ | | WC1.2$^+$ | |
|---|---|---|---|---|---|
| Name | Symbol | Log$_2$FC | FDR-$P_{adj.}$ | Log$_2$FC | FDR-$P_{adj.}$ |
| Toll-like receptor 4 | TLR4 | 1.7 | $1.08 \times 10^{-12}$ | 1.87 | $3.49 \times 10^{-15}$ |
| Pentraxin 3 | PTX3 | 3.74 | $2.81 \times 10^{-8}$ | 4.6 | $6.03 \times 10^{-12}$ |
| Interferon Regulatory Factor 1 | IRF1 | 3.41 | $3.19 \times 10^{-7}$ | 3.35 | $1.18 \times 10^{-45}$ |
| Interferon Regulatory Factor 4 | IRF4 | 2.14 | $5.28 \times 10^{-8}$ | 3.33 | $8.85 \times 10^{-19}$ |
| Nuclear factor kappa-light-chain-enhancer of activated B cells 2 | NFkB2 | 1.69 | $2.74 \times 10^{-7}$ | 1.88 | $7.11 \times 10^{-9}$ |
| Interleukin 1α | IL1A | 4.11 | $5.97 \times 10^{-35}$ | 4.21 | $2.6 \times 10^{-6}$ |
| Interleukin 1β | IL1B | 3.48 | $1.43 \times 10^{-5}$ | 4.07 | $3.53 \times 10^{-7}$ |
| Interleukin 2 | IL2 | 4.51 | $2.28 \times 10^{-5}$ | 4.96 | $4.71 \times 10^{-6}$ |
| Interleukin 6 | IL6 | 3.22 | 0.00037 | 4.19 | $9.15 \times 10^{-6}$ |
| Serum Amyloid A3 | SAA3 | 4.98 | $6.44 \times 10^{-6}$ | 4.83 | $1.12 \times 10^{-5}$ |
| Interleukin 17α | IL17A | 1.68 | $2.27 \times 10^{-6}$ | 2.23 | 0.004 |
| Interleukin 17F | IL17F | 4.25 | $2.98 \times 10^{-6}$ | 2.31 | 0.004 |
| Interferon-gamma | IFNG | 3.90 | $7.24 \times 10^{-5}$ | 5.61 | $9.41 \times 10^{-8}$ |
| Leukemia Inhibitory Factor | LIF | 4.75 | $6.44 \times 10^{-11}$ | 5.9 | $4.86 \times 10^{-9}$ |
| C-X-C Motif Chemokine Ligand 10 | CXCL10 | 6.95 | $6.35 \times 10^{-46}$ | 7.56 | $5.01 \times 10^{-46}$ |
| C-X-C Motif Chemokine Ligand 8 | CXCL8 | 2.96 | $1.04 \times 10^{-5}$ | 3.36 | $6.6 \times 10^{-7}$ |
| Vitamin D Receptor | VDR | 2.5 | $5.63 \times 10^{-9}$ | 2.4 | $3.65 \times 10^{-8}$ |

FDR-$P_{adj.}$ = $5.28 \times 10^{-8}$; log$_2$FC in WC1.2$^+$ = 3.33, FDR-$P_{adj.}$ = $8.95 \times 10^{-19}$) on the other hand has roles in IL-17 production via Th17 cells differentiation, which is also critical for containing mycobacteria [32]. Similarly, *NFkB2* in WC1.1$^+$ (log$_2$FC = 1.69, FDR-$P_{adj.}$ = $2.74 \times 10^{-7}$) and in WC1.2$^+$ (log$_2$FC = 1.88, FDR-$P_{adj.}$ = $7.11 \times 10^{-9}$) is involved in the NF-κB pathway, and important in regulating the expression of numerous inflammatory cytokines and chemokines, indicating activation of a robust inflammatory response [33].

The cytokine profiles (*IL1A*, *IL1B*, *IL2*, *IL6*, *IL17A*, and *IL17F*) were all upregulated in both WC1.1$^+$ and WC1.2$^+$ cells. A previous study showed that IL-6, IL-12, IP-10 and IFN-γ were significantly higher when PBMCs from *M. bovis*-infected cattle were stimulated with PPDb [34]. *IL1A* expression was upregulated by a log$_2$FC of 4.11 with FDR-$P_{adj.}$ = $5.97 \times 10^{-35}$ in WC1.1$^+$ and a log$_2$FC of 4.21 with FDR-$P_{adj.}$ = $2.6 \times 10^{-6}$ in WC1.2$^+$ γδ T cell subsets. The expression of *IL1B* (log$_2$FC = 4.06, FDR-$P_{adj.}$ = $3.52 \times 10^{-7}$) and *IL6* (log$_2$FC = 4.19, FDR-$P_{adj.}$ = $9.14 \times 10^{-6}$) in WC1.2$^+$ were slightly higher than in WC1.1$^+$ cells (log$_2$FC = 3.49, FDR-$P_{adj.}$ = $1.43 \times 10^{-5}$). Another upregulated cytokine-like protein, *SAA3* (log$_2$FC = 4.98, FDR-$P_{adj.}$ = $2.59 \times 10^{-5}$ in WC1.1$^+$; log$_2$FC = 4.83, FDR-$P_{adj.}$ = $1.12 \times 10^{-5}$ in WC1.2$^+$) is demonstrated to induced IL-1β production in macrophages stimulated with *M. tuberculosis* in an ESAT-6 dependent manner [35]. *IL2* also showed substantial upregulation both in WC1.2$^+$ cells (log$_2$FC = 4.96, FDR-$P_{adj.}$ = $4.71 \times 10^{-6}$) and WC1.1$^+$ cells (log$_2$FC = 4.51, FDR-$P_{adj.}$ = $2.28 \times 10^{-5}$). IL-2 regulates the differentiation of T cells and may reverse T cell dysfunction induced by persistent *M. tuberculosis* infection [36].

An increase in the expression of *IL6* (log$_2$FC = 4.19, FDR-$P_{adj.}$ = = $9.14 \times 10^{-6}$) and *LIF* (log$_2$FC = 5.9, FDR-$P_{adj.}$ = $4.86 \times 10^{-9}$) was observed in the stimulated WC1.2$^+$ subset. In the WC1.1$^+$ subset, *IL6* showed a log$_2$FC of 3.22 with an FDR-$P_{adj.}$ of $3.67 \times 10^{-4}$, and *LIF* exhibited a log$_2$FC of 4.75 with an FDR-$P_{adj.}$ of $6.44 \times 10^{-11}$. IL-6 and LIF are members of the IL-6 cytokine family, primarily recognized as pro-inflammatory cytokines, though they also exhibit regulatory functions [37]. The production of IL-6 is influenced by *M. tuberculosis*, which impairs IFN-γ signalling in host macrophages and subsequently promotes disease progression

[38]. In humans, levels of LIF have been shown to be significantly higher in both active and latent tuberculosis compared to non-TB patients, suggesting its potential role in the regulation of inflammation [39].

The downstream signalling upon stimulation with mycobacterial antigen also activates chemokine production, induces the recruitment of immune cells to sites of infection, and is important in regulating inflammation. The chemokines *CXCL8* (IL-8) and *CXCL10* (IP-10) are upregulated in both WC1.1$^+$ and WC1.2$^+$ γδ T cells in response to PPDb. *CXCL8* (log$_2$FC in WC1.1$^+$ = 2.96, FDR-$P_{adj.}$ = 1.04 × 10$^{-5}$; log$_2$FC in WC1.2$^+$ = 3.36, FDR-$P_{adj.}$ = 6.59 × 10$^{-7}$) is a potent chemoattractant for neutrophils [16–19, 40], and C*XCL10* (log$_2$FC in WC1.1$^+$ = 6.95 with FDR-$P_{adj.}$ = 6.35 × 10$^{-46}$; log$_2$FC in WC1.2$^+$ = 7.56 with FDR-$P_{adj.}$ = 5 × 10$^{-46}$) recruits activated T cells to the site of infection, highlighting the suggestive role of γδ T cells in coordinating the immune cell infiltration [41]. Interestingly, the expression of *CXCL10* has been recently proposed to improve the diagnosis of *M. tuberculosis* infection [42].

γδ T cells play a pivotal role as an early and crucial source of IFN-γ, which is critical in promoting T helper 1 type immunity during infection [43]. WC1.1$^+$ and WC1.2$^+$ γδ T cell subsets showed substantial upregulation of *IFNG*, where log$_2$FC in WC1.1$^+$ is 3.90 with FDR-$P_{adj.}$ = 7.24 × 10$^{-5}$ and log$_2$FC in WC1.2$^+$ is 5.61 with FDR-$P_{adj.}$ = 9.41 × 10$^{-8}$). McGill *et al.* [17] also found a more specific and direct response of WC1$^+$ γδ T cells to mycobacterial antigens, including PPDb. Interestingly, our study showed higher IFN-γ expression in WC1.2$^+$ cells, contrary to earlier reports where WC1.1$^+$ cells exhibited a greater response when stimulated with mycobacterial antigens [16–19]. Furthermore, both subsets showed elevated expression of IL-17A and IL-17F, key cytokines in the recruitment of neutrophils and the formation of granuloma during *M. tuberculosis* infection [44]. Notably, *IL17F* had higher expression in WC1.1$^+$ cells (log$_2$FC = 4.24, FDR-$P_{adj.}$ = 2.98 × 10$^{-6}$) than WC1.2$^+$ cells (log$_2$FC = 2.23, FDR-$P_{adj.}$ = 0.0043. Despite WC1.1$^+$ showing a stronger response to PPDb stimulation for IL-17F, both subsets had similar levels of expression of IL17A. This is supported by Lockhart *et al.* [45], who also showed γδ T cells as the primary source of IL-17 production rather than CD4 T cells during *M. tuberculosis* infection. By producing pro-inflammatory cytokines such as IFN-γ, TNF-α, and IL-17 in response to mycobacterial antigen, WC1$^+$ γδ T cell subsets contribute to the activation of antigen-presenting cells and other effector immune cells, thereby enhancing the clearance of infection [46].

The differential expression of genes implicated in regulating vitamin D metabolism and signalling, notably VDR (log$_2$FC in WC1.1$^+$ = 2.46 with FDR-$P_{adj.}$ = 5.64 × 10$^{-9}$; log$_2$FC in WC1.2$^+$ = 2.35 with FDR-$P_{adj.}$ = 3.66 × 10$^{-8}$) in both subsets, was also observed. A previous study using macrophages from recovered persons from extrapulmonary tuberculosis also showed increased expression of VDR after stimulation in *M. tuberculosis* [47].

Overall, the upregulation of *IL1A*, *IL1B*, *IL6*, *LIF*, and *IFNG* suggests a strong pro-inflammatory cytokine response to PPDb, WC1.2$^+$ cells from bTB+ cattle responding slightly stronger than WC1.1$^+$ to mycobacterial antigen. These findings are expected components of an inflammatory response and agree with a previous study that showed that IL-6, IL-12, IP-10 and IFN-γ were significantly higher when PBMCs from *M. bovis*-infected cattle were stimulated with PPDb [34].

## Gene and pathway enrichment analysis of DEGs from PPDb stimulated WC1.1$^+$ and WC1.2$^+$ γδ T cells

A Gene Ontology (GO) term enrichment analysis elucidated that WC1.1$^+$ and WC1.2$^+$ cells exhibited significant enrichment in immune-related processes, cytokine regulation, cellular responses to stimuli, and defense mechanisms in the cellular response to PPDb (Fig 4). Both cell subsets showed enrichment in "*Immune Response*" (GO:0006955) and "*Immune System Process*" (GO:0002376).

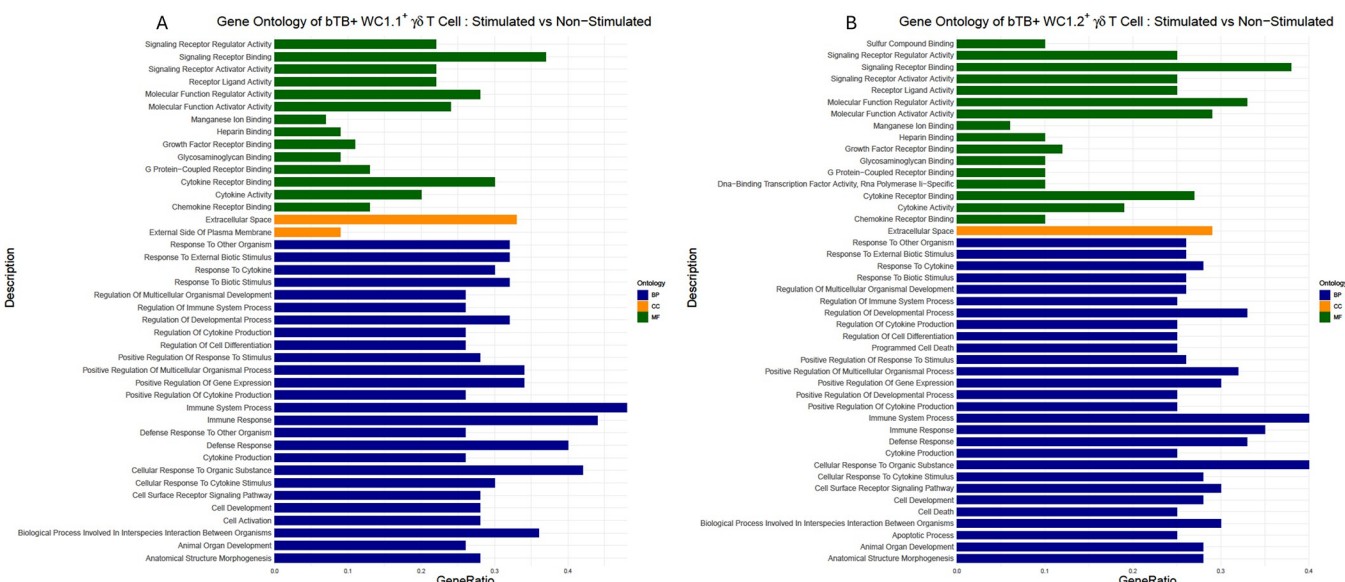

**Fig 4.** GO term enrichment analysis of DEGs by RNA sequencing for the (A) WC1.1+ and (B) WC1.2+ γδ T cell subsets from bTB+ cattle (*n* = 5) in response to a 6-hour stimulation with 10 μg/ml PPDb compared to the non-stimulated. The vertical coordinates are the enriched GO terms, and the horizontal coordinates are the numbers of the DEGs in these GO terms. The dark blue columns represent the biological process GO terms; the dark orange columns represent the cellular component GO terms; the dark green columns represent the molecular function GO terms.

Additionally, terms related to cytokine regulation, such as "*Positive Regulation of Cytokine Production*" (GO:0001819) and "*Response to Cytokine*" (GO:0034097), were prominent in both subsets. Both WC1.1+ and WC1.2+ cells demonstrated enrichment in "*Cellular Response to Cytokine Stimulus*" (GO:0071345), "*Defense Response*" (GO:0006952) and "*Cellular Response to Organic Substance*" (GO:0071310), indicating their ability to sense and respond to extracellular signals. The significant enrichment of similar GO terms in WC1.1+ and WC1.2+ cells indicate common pathways and processes underlying their responses to PPDb stimulation are similarly activated.

A Kyoto Encyclopedia of Genes and Genomes (KEGG) pathway-enriched analysis in WC1.1+ and WC1.2+ cells following PPDb stimulation revealed shared pathways, indicating conserved mechanisms in their responses (Fig 5). Both cell types exhibited enrichment in pathways such as cytokine-cytokine receptor interaction (bta04060), enrichment in the viral protein interaction pathway (bta04061) and enrichment in TNF signalling (bta04668). Transcriptomic results from both cell types also showed enrichment in IL-17 signalling (bta04657) and inflammatory bowel disease (bta05321) pathways.

The enrichment of NF-kappa B signalling (bta04064) and enrichment of JAK-STAT signalling (bta04630) suggests involvement in mediating cytokine and inflammatory responses. Finally, involvement in the NOD-like receptor signalling pathway (bta04621) suggests roles in innate immune responses to pathogens. The shared pathways between the two cell types again reflect the overall conserved response pathways to PPDb stimulation in γδ T cell subsets from *M. bovis* infected cattle.

## Limited subset-specific differential gene expression in response to PPDb stimulation

Although the majority of genes (111) were differentially expressed in both subsets in response to PPDb and similar pathways were significantly enriched (as discussed above), a small

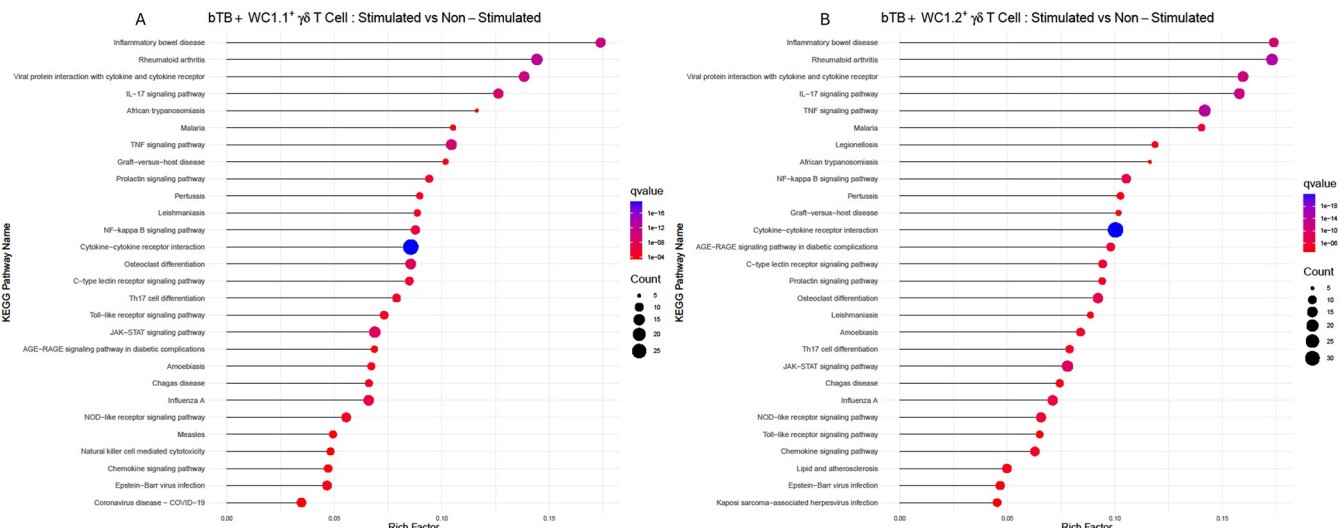

**Fig 5. Statistics of the Kyoto Encyclopedia of Genes and Genomes (KEGG) pathway enrichment analysis of DEGs by RNA-seq for the γδ T cell subsets from bTB+ cattle (*n* = 5) in response to a 6-hour stimulation with 10 μg/ml PPDb compared to the non-stimulated (A) WC1.1+ and (B) WC1.2+.** The x-axis represents the percentage of enriched genes to the total annotated genes. The y-axis indicates the description of each enrichment category. The size of the dots corresponds to the number of enriched genes, and the colour panel on the right indicates the q value, with the lower values corresponding to higher significance.

number of genes reached the statistical significance threshold of $|\log_2 FC| \geq 1.5$ and FDR-$P_{adj.}$ < 0.1 in one or other γδ T cells subset only, and which may reflect a preferential ability of one subset to activate immune relevant responses. The WC1 co-receptor is thought to be an important determining element in the activation process and subsequent response [48], and previous research has elucidated that both WC1.1+ and WC1.2+ cell subsets are known to differentially respond to mycobacterial antigens [16–19, 49].

In this study, 14 genes in the WC1.1+ subset and 43 genes in the WC1.2+ subset met the threshold cut-off and are listed in S1 Table. Selected genes with well-established roles in the immune response are shown in Table 2.

The differential gene expression analysis also highlights genes with higher expression in the WC1.1+ subset, including *ACKR3, RTP4,* and *GBP5*. These genes encompass diverse functions, from immune cell signalling to chemotaxis and lipid metabolism [50]. Guanylate-binding protein 5 (GBP5) is a GTPase highly inducible by interferon that promotes inflammasome assembly [51] which is critical for inflammatory protein cleavage and activation, including in response to mycobacteria [52]. *LOC507055* encodes guanylate-binding protein 4 (GBP4), another member of the same family and is also differentially expressed here. A*CKR3*, known as *CXCR7* ($\log_2 FC$ = 1.52, FDR-$P_{adj.}$ = $7.87 \times 10^{-15}$), likely modulates immune cell migration [53]. These genes collectively underscore the unique functional characteristics of WC1.1+ γδ T cells and suggest potential roles in immune surveillance and response.

Furthermore, the WC1.2+ subset displayed upregulation of immune modulators. This included *CCL2* ($\log_2 FC$ = 1.73, FDR-$P_{adj.}$ = 0.009) and *CCL22* ($\log_2 FC$ = 2.17, FDR-$P_{adj.}$ = 0.004), which facilitate the recruitment of monocytes, memory T cells, dendritic cells, and regulatory T cells (Tregs), indicating that WC1.2+ cells are initiating and sustaining mycobacterial antigen response while also modulating immune homeostasis to prevent excessive inflammation [54, 55]. *CXCL2* ($\log_2 FC$ = 2.28, FDR-$P_{adj.}$ = 0.002) and *CXCL3* ($\log_2 FC$ = 3.22, FDR-$P_{adj.}$ = 0.0004) regulate lipid metabolism by interacting with lipid-related genes in TB infections, which may be essential for the initial defense against infections [56, 57]. Additionally, the

**Table 2. Genes exhibiting statistically significant expression (FDR-$P_{adj.}$ < 0.10; |log$_2$FC| ≥ 1.5) increased expression in either WC1.1$^+$ or WC1.2$^+$ γδ T cell subsets from bTB+ cattle (n = 5) in response to a 6-hour stimulation with 10 μg/ml PPDb compared to the non-stimulated control.**

| Gene Name | Gene Symbol | Log$_2$FC | FDR-$P_{adj.}$ | γδ T Cell Subset expression above cut-off threshold |
|---|---|---|---|---|
| C-X-C Motif Chemokine Receptor 7, also known as Atypical Chemokine Receptor 3 | CXCR7 or ACKR3 | 1.52 | $7.87 \times 10^{-15}$ | WC1.1$^+$ |
| C-X-C Motif Chemokine Ligand 9 | CXCL9 | 3.23 | 0.0009 | WC1.1$^+$ |
| Guanylate Binding Protein 5 | GBP5 | 1.73 | 0.0003 | WC1.1$^+$ |
| Fatty Acid Binding Protein 5 | FABP5 | 1.5 | 0.0008 | WC1.1$^+$ |
| C-C Motif Chemokine Ligand 2 | CCL2 | 1.73 | 0.009 | WC1.2$^+$ |
| C-C Motif Chemokine Ligand 22 | CCL22 | 2.17 | 0.004 | WC1.2$^+$ |
| C-X-C Motif Chemokine Ligand 2 | CXCL2 | 2.28 | 0.002 | WC1.2$^+$ |
| C-X-C Motif Chemokine Ligand 3 | CXCL3 | 3.22 | 0.0004 | WC1.2$^+$ |
| Interleukin 17 Receptor E-Like | IL17REL | 2.27 | 0.001 | WC1.2$^+$ |
| Interleukin 1 Receptor Antagonist | IL1RN | 1.54 | 0.0004 | WC1.2$^+$ |
| Granzyme A | GZMA | 1.99 | 0.007 | WC1.2$^+$ |

expression of *GZMA* (log$_2$FC = 1.99, FDR-$P_{adj.}$ = 0.007) in WC1.2$^+$ γδ T cells highlights their cytotoxic capabilities to kill infected or transformed cells directly, thus controlling infections and eliminating potentially malignant cells [24, 58]. Higher transcript patterns of *CCL2*, *CCL22*, *CXCL2*, and *CXCL3* indicate that WC1.2$^+$ cells might be more actively involved in chemotaxis and immune response compared to WC1.1$^+$ cells, based on the higher fold changes and significant values. Other published work has shown that *M. bovis*-responsive bovine γδ T cells expressed higher surface levels of CXCR3 compared to non-responding γδ T cells, a feature proposed to enable them to migrate to inflamed tissues [59].

## Direct comparison between PPDb stimulated WC1.1$^+$ and WC1.2$^+$ expression profiles

Results presented thus far suggest that the transcriptomic response to PPDb stimulation is largely conserved between γδ T cell subsets. However, as documented in previous work, WC1.1$^+$ γδ T cells from cattle naturally infected with *M. bovis*-infected are more highly activated compared to control groups [24], and therefore, basal activation status (prior to PPDb stimulation) is not identical. Therefore, in this study, a direct comparison between both stimulated subsets after 6 h stimulation with PPDb was performed. Using the same stringent thresholds (|log$_2$FC| ≥ 1.5 and FDR-$P_{adj.}$ < 0.10), a total of 13 genes were differentially expressed, 12 of which were significantly increased in the WC1.1$^+$ γδ T cell subset (Table 3 and S4 Table).

Similarly, the upregulation of *MS4A1* (also known as *CD20*) (log$_2$FC of 1.62, FDR-$P_{adj.}$ = $9.41 \times 10^{-24}$) and *CD22* (log$_2$FC of 1.67, FDR-$P_{adj.}$ = 0.004) in stimulated WC1.2$^+$ cells compared to stimulated WC1.1$^+$ also highlights the potential involvement of these γδ T cell subsets in B-cell-mediated immunity [60, 61]. CD20$^+$ T cells are also an emerging subset of traditional αβ T cells in human pathology, including infection, cancer and autoimmune diseases [62]. Our data suggests that CD20 expression is not exclusive to traditional αβ T cells or B cells, highlighting the diverse and complex nature of the γδ T cell subsets in response to mycobacterial stimulation [63]. CD22 is a sugar binding transmembrane protein, which specifically binds sialic acid with an immunoglobulin (Ig) and thereby also regulates B cell responses [60]. This is of relevance in the context of the emerging evidence supporting interactions between γδ T and B cells [64], and B cells have an underappreciated role in the context of TB infection [65]. WC1 is increased in expression in the WC1.1$^+$ cells and a minority fraction of this γδ T cell subset are reported to also express another WC1 allele known as WC1.3 on their surface

**Table 3. Genes exhibiting statistically significant expression (FDR-$P_{\text{adj.}}$ < 0.10; |log$_2$FC| ≥ 1.5) between WC1.1$^+$ and WC1.2$^+$ γδ T cell subsets from bTB+ cattle ($n$ = 5) after a 6-hour stimulation with 10 μg/ml PPDb.**

| Gene Name | Gene Symbol | Log$_2$FC | FDR-$P_{\text{adj.}}$ |
|---|---|---|---|
| Workshop Cluster 1.3 | WC1.3 | -2.98 | $9.41 \times 10^{-24}$ |
| Neuronal Cell Adhesion Molecule | NRCAM | 1.52 | 0.02 |
| Epoxide Hydrolase 4 | EPHX4 | 1.56 | 0.004 |
| Membrane Spanning 4-Domain A1, also known as Cluster of Differentiation-20 | MS4A1 or CD20 | 1.62 | 0.002 |
| Fc Receptor-Like 1 | FCRL1 | 1.63 | 0.0006 |
| Cluster of Differentiation-22 | CD22 | 1.67 | 0.004 |
| Solute Carrier Family 6 Member 17 | SLC6A17 | 1.72 | 0.01 |
| Leucine Rich Repeat Containing G Protein-Coupled Receptor 4 | LGR4 | 1.73 | 0.006 |
| Kelch Domain Containing 8A | KLHDC8A | 1.81 | 0.002 |
| Kelch-Like Family Member 14 | KLHL14 | 1.93 | 0.002 |
| Deltex E3 Ubiquitin Ligase 4 | DTX4 | 1.97 | $3.52 \times 10^{-5}$ |
| Workshop Cluster 1 | WC1 | 2.05 | $6.33 \times 10^{-7}$ |
| Sodium Voltage-Gated Channel Alpha Subunit 3 | SCN3A | 2.21 | $6.33 \times 10^{-7}$ |

[49]. However, the expression of WC1.3 was downregulated by a log$_2$FC of 2.98 with FDR-$P_{adj.}$ = $9.41 \times 10^{-24}$ in stimulated WC1.1$^+$ cells when compared with stimulated WC1.2$^+$.

## Conclusion

The eradication of bTB is a major policy aim in countries where the disease is endemic, as it has widespread negative impacts on the agri-food sector and poses a zoonotic risk to human health [2, 3]. The γδ T cell subpopulation has attracted significant research attention across species due to their developmental and functional plasticity. Although transcriptomics approaches have been used to interrogate the effects of both coding and non-coding transcript changes [66, 67] on cell function, a lot remains to be discovered [9, 68], particularly in livestock and non-model organisms. γδ T cells are potent immune cells that are expanded in cattle and have many critical effector functions. Their role in protective immunity is obviously of critical relevance to many infections, but as producers of sentinel cytokines on which CMI-based diagnostic tests for bTB rely, their influence on other effector cell subsets warrants particular attention [69].

The WC1 co-receptor is an important determining element in the activation process and subsequent response [48], and here, we document a shared inflammatory profile of both WC1.1$^+$ and WC1.2$^+$ γδ T cell compartments in response to mycobacterial restimulation with bovine PPD, showing the capacity of both subsets to produce classical effector cytokines, including *IL1A*, *IL1B*, *IL17A*, *IL17F*, and *IFNG*. However, some evidence for subset-specific responses was also apparent, with significant upregulation of *IRF7* detected only in the WC1.1$^+$ cells; the WC1.2$^+$ cells showed upregulation of *CCL22* and *GZMA*, which encodes the cytotoxic Granzyme A. Differences in anti-inflammatory and antimicrobial gene expression between these cell subsets provide new insights into their effector functions in response to mycobacterial antigens.

Other studies have used alternative markers to assess bovine γδ T cell function, noting, for example, that half of γδ T cells shown responsive to PPDb (and other mycobacterial antigens) expressed CD27 receptor as measured by flow cytometry [59]. CD27$^+$ *M. bovis*-responsive γδ T cells also co-expressed CD62L compared to γδ T cells that did not proliferate to *M. bovis* antigens, and the same study showed that CD45RO correlated with memory-type responses. In this study, neither CD45, CD62L, nor CD27 were found to be significantly differentially

expressed, which is likely a feature of the different time course investigated, and the cocktail of mycobacterial antigens used (ESAT-6) used in other work. Furthermore, it is likely that results in naturally infected cattle will be different to immune responses in experimentally infected cattle as detected in the previously discussed study. Another previous study suggested that WC1.1[+] and WC1.2[+] γδ T cell subsets differ in terms of TGFb production [49], and this cytokine was significantly differentially expressed in the WC1.2[+] γδ subset in this study, but the fold change was marginal. The results presented here add weight to the functional plasticity of WC1[+] γδ T cell subsets in cattle, as has been documented in other species [70], and further confirmatory research in additional samples and potentially incorporating the use of single cell sequencing is needed to more fully elucidate the specific functions of these critically important γδ T cells under various disease conditions in livestock.

## Supporting information

**S1 Fig. Volcano plot of differentially expressed genes (DEGs) in WC1.1[+] γδ T cell subset in response to a 6-hour stimulation with 10 μg/ml PPDb compared with WC1.2[+] γδ T cell subset from bTB+ cattle (n = 5).** Red dots represent both upregulated and downregulated genes with $|\log_2 FC| \geq 1.5$ and FDR-$P_{adj.} < 0.10$ (above the black dashed lines). Grey dots indicate DEGs with FDR-$P_{adj.} < 0.10$ but $|\log_2 FC| < 1.5$. Black dots represent non-significant genes below the threshold of $|\log_2 FC| \geq 1.5$ and FDR-$P_{adj.} < 0.10$.
(PDF)

**S1 Table. Full list of Differentially Expressed Genes (DEGs) in WC1.1[+] γδ T cell subset.** WC1.1[+] γδ T cells from bTB+ cattle (*n* = 5) in response to a 6-hour stimulation with 10 μg/ml PPDb compared to the non-stimulated control. The Content Sheet in the S1 Table represents all the summary information.
(XLSX)

**S2 Table. Full list of Differentially Expressed Genes (DEGs) in WC1.2[+] γδ T cell subset.** WC1.2[+] γδ T cells from bTB+ cattle (*n* = 5) in response to a 6-hour stimulation with 10 μg/ml PPDb compared to the non-stimulated control. The Content Sheet in the S2 Table represents all the summary information.
(XLSX)

**S3 Table. Overlapping DEGs in both bTB+ WC1.1[+] stimulated with PPDb vs non-stimulated (Group 1) and WC1.2[+] stimulated with PPDb vs non-stimulated (Group 2) γδ T cell subsets.** The Content Sheet in the S3 Table represents all the summary information.
(XLSX)

**S4 Table. Full list of Differentially Expressed Genes (DEGs) in response to PPDb in WC1.1[+] and WC1.2[+] γδ T cell subsets.** WC1.1[+] γδ T cells from bTB+ cattle (*n* = 5) in response to a 6-hour stimulation with 10 μg/ml PPDb compared to the WC1.2[+] γδ T cells from bTB+ cattle (*n* = 5) in response to a 6-hour stimulation with 10 μg/ml PPDb. The Content Sheet in the S4 Table represents all the summary information.
(XLSX)

## Author Contributions

**Conceptualization:** Kieran G. Meade.

**Formal analysis:** Alia Parveen, Mahmoud Elnaggar, Kieran G. Meade.

**Funding acquisition:** Kieran G. Meade.

**Investigation:** Mahmoud Elnaggar, Kieran G. Meade.

**Methodology:** Sajad A. Bhat, Mahmoud Elnaggar.

**Project administration:** Sajad A. Bhat.

**Supervision:** Kieran G. Meade.

**Writing – original draft:** Alia Parveen, Kieran G. Meade.

**Writing – review & editing:** Alia Parveen, Mahmoud Elnaggar, Kieran G. Meade.

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
