## [Decision Letter · Decision Letter 0]

6 Sep 2024

PONE-D-24-33992Comparative analysis of WC1.1+ and WC1.2+ γδ T cell subset responses from cattle naturally infected with Mycobacterium bovis to repeat stimulation with mycobacterial antigens.PLOS ONE

Dear Dr. Meade,

Thank you for submitting your manuscript to PLOS ONE. After careful consideration, we feel that it has merit but does not fully meet PLOS ONE’s publication criteria as it currently stands. Therefore, we invite you to submit a revised version of the manuscript that addresses the points raised during the review process.

We look forward to receiving your revised manuscript.

Kind regards,

Yung-Fu Chang

Academic Editor

PLOS ONE

Journal Requirements:

10.1186/s12864-024-10574-x

In your revision ensure you cite all your sources (including your own works), and quote or rephrase any duplicated text outside the methods section. Further consideration is dependent on these concerns being addressed.

3. Please expand the acronym “HEA” (as indicated in your financial disclosure) so that it states the name of your funders in full.

"This project was funded by Science Foundation Ireland (SFI) Grant 17/CDA/4717 and AP was funded by HEA Vit-TB."

5. Please note that funding information should not appear in the Acknowledgments section or other areas of your manuscript. We will only publish funding information present in the Funding Statement section of the online submission form. Please remove any funding-related text from the manuscript. 

**Additional Editor Comments:**

Your manuscript has been reviewed by an expert in your field. Based on the comments, a minor revision is needed.

Reviewers' comments:

Reviewer's Responses to Questions

**Comments to the Author**

1. Is the manuscript technically sound, and do the data support the conclusions?

Reviewer #1: Yes

2. Has the statistical analysis been performed appropriately and rigorously? 

Reviewer #1: Yes

3. Have the authors made all data underlying the findings in their manuscript fully available?

Reviewer #1: Yes

4. Is the manuscript presented in an intelligible fashion and written in standard English?

Reviewer #1: Yes

5. Review Comments to the Author

Reviewer #1: The work “Comparative analysis of WC1.1+ and WC1.2+ γδ T cell subset responses from cattle naturally infected with Mycobacterium bovis to repeat stimulation with mycobacterial antigens” report some interesting findings in specific T cell populations from natural tb-infected cows.

Although I think the manuscript is well written and straightforward, I found that the lack of line numbers makes it difficult to point out errors and revision requests.

Here are some comments to be considered by the authors before publication.

General comments on manuscript:

- The use of bulk RNAseq strategy to characterize γδ T cells response to PPD is still a useful analysis method to explore coding transcriptome, and interesting data can therefore be collected by this approach. However, considering currently available single-cell RNAseq technologies, I suggest the author to revise the conclusion sections of their manuscript and to add some comments in this regard.

- Moreover, in order to further enrich the discussion of their paper, I invite the authors also to report and cite some studies regarding non-coding transcriptome (lncRNA.miRNA,etc..) that seems to modulate specific gene expression pathways related to M. bovis infection.

- How the fixed time course of the PPD exposure (6 hours) was chosen by the authors? Did the authors consider also other PPD incubation times?

- Regarding recorded data, I think a confirmatory analysis by qPCR (Real Time PCR, digital PCR) on most discriminant genes for both WC1.1+ and WC1.2+ is still useful and should be considered for validation of DEG found by sequencing analysis. This aspect is crucial also in order to extend in the future their findings to supplementary field samples without the use of costly RNAseq. Why the author did not include a confirmatory gene expression study in their study? I invite them to justify their choices in the result/discussion sections.

Specific comments to manuscript sections:

- Please check through the text the use of italics for scientific names (for example B. taurus in the section RNA-seq Data Processing and Analysis).

- In the section GO and KEGG Enrichment Analyses please change this sentence “Lists of genes with increased |log2FC| ≥ 1.5 and FDR-Padj. < 0.10) or decreased (|log2FC| ≥ 1.5, and FDR-Padj. < 0.10)..” with “Lists of genes with increased or decreased expression (|log2FC| ≥ 1.5 andFDR-Padj. < 0.10)…”

6. PLOS authors have the option to publish the peer review history of their article (what does this mean?). If published, this will include your full peer review and any attached files.

Reviewer #1: No

---

## [Decision Letter · Decision Letter 1]

25 Sep 2024

Comparative analysis of WC1.1+ and WC1.2+ γδ T cell subset responses from cattle naturally infected with Mycobacterium bovis to repeat stimulation with mycobacterial antigens.

PONE-D-24-33992R1

Dear Dr. Meade,

We’re pleased to inform you that your manuscript has been judged scientifically suitable for publication and will be formally accepted for publication once it meets all outstanding technical requirements.

Kind regards,

Yung-Fu Chang

Academic Editor

PLOS ONE

Additional Editor Comments (optional):

Reviewers' comments:

Reviewer's Responses to Questions

**Comments to the Author**

1. If the authors have adequately addressed your comments raised in a previous round of review and you feel that this manuscript is now acceptable for publication, you may indicate that here to bypass the “Comments to the Author” section, enter your conflict of interest statement in the “Confidential to Editor” section, and submit your "Accept" recommendation.

Reviewer #1: All comments have been addressed

2. Is the manuscript technically sound, and do the data support the conclusions?

Reviewer #1: Yes

3. Has the statistical analysis been performed appropriately and rigorously? 

Reviewer #1: Yes

4. Have the authors made all data underlying the findings in their manuscript fully available?

Reviewer #1: Yes

5. Is the manuscript presented in an intelligible fashion and written in standard English?

Reviewer #1: Yes

6. Review Comments to the Author

Reviewer #1: (No Response)

7. PLOS authors have the option to publish the peer review history of their article (what does this mean?). If published, this will include your full peer review and any attached files.

Reviewer #1: No

---

## [Editor Report · Acceptance letter]

1 Oct 2024

PONE-D-24-33992R1 

PLOS ONE

Dear Dr. Meade, 

I'm pleased to inform you that your manuscript has been deemed suitable for publication in PLOS ONE. Congratulations! Your manuscript is now being handed over to our production team.

Kind regards, 

on behalf of

Dr. Yung-Fu Chang 

Academic Editor

PLOS ONE